

# Land use affects the response of soil moisture and soil temperature to environmental factors in the loess hilly region of China

Min Tang[1], Wanning Li[1], Xiaodong Gao[2], Pute Wu[2], Hongchen Li[3], Qiang Ling[4] and Chao Zhang[1]

[1] College of Hydraulic Science and Engineering, Yangzhou University, Yangzhou, Jiangsu, China
[2] Institute of Water-saving Agriculture in Arid Areas of China, Northwest Agriculture and Forestry University, Yangling, Shaanxi, China
[3] School of Resources and Environmental Engineering, Ludong University, Yantai, Shandong, China
[4] College of Environmental and Resource Sciences, Zhejiang A&F University, Hangzhou, Zhejiang, China

Corresponding author
Chao Zhang, zhangc1700@yzu.edu.cn

## ABSTRACT

Changes in soil moisture and soil temperature result from the combined effects of several environmental factors. Scientific determination of the response characteristics of soil moisture and soil temperature to environmental factors is critical for adjusting the sloping land use structure and improving the ecological environment in China's loess hilly region. Soybean sloping fields, maize terraced fields, jujube orchards, and grasslands in the loess hilly region were selected as the research areas. The change in characteristics of soil moisture and soil temperature, as well as their interactions and statistical relationships with meteorological factors, were analyzed using continuously measured soil moisture, soil temperature, and meteorological factors. The results revealed that air temperature and humidity were the main controlling factors affecting soil moisture changes in the 0–60 cm soil layer of soybean sloping fields and grasslands in the normal precipitation year (2014) and the dry year (2015). Humidity and wind speed were the main meteorological factors affecting soil moisture changes in the maize terraced field. Air temperature had a significant negative effect on soil moisture in the jujube orchard. Soil moisture and soil temperature were all negatively correlated under the four sloping land use types. In normal precipitation years, atmospheric humidity had the greatest direct and comprehensive effect on soil moisture in soybean sloping fields, maize terraced fields, and grasslands; soil temperature had a relatively large impact on soil moisture in jujube orchards. The direct and comprehensive effects of soil temperature on soil moisture under all sloping land use types were the largest and most negative in the dry year. Air temperature had a high correlation with soil temperature in the 0–60 cm soil layer under the four sloping land use types, and the grey relational grade decreased as the soil layer deepened. The coefficient of determination between the 0–20 cm soil temperature and air temperature in the maize terraced field was low, indicating a weak response to air temperature. The above findings can serve as a scientific foundation for optimizing sloping land use structures and maximizing the efficient and sustainable utilization of sloping land resources in China's loess hilly region.

## INTRODUCTION

Soil moisture distribution and variability are reflections of and are determined by the interactions between several environmental factors (that might be impacted by human activity) such as climate, vegetation, topography, and soil properties (*Zhu et al., 2014*; *Huang et al., 2016*). Soil moisture variability is primarily controlled by a variety of meteorological factors on a time scale (*Li et al., 2019*; *Zhu et al., 2019*). For instance, changes in air temperature can influence not only the energy state and availability of soil moisture but also the retention, diffusion capacity, and loss of soil moisture (*Pérez, 1998*). Precipitation can increase relative humidity in the air and soil moisture content (*Chen et al., 2009b*). Other meteorological factors, such as wind speed, solar radiation, and water vapor pressure, can also affect the evapotranspiration of soil moisture (*Messing et al., 1998*). *Zhong et al. (2014)* found that, in the hilly area of Chongqing, China, temperature, sunshine, and precipitation are inversely correlated with the temporal variation of soil moisture in the 0–10 cm layer from March to May and temperature and sunshine are inversely correlated with the temporal variation of soil moisture in the 0–10 cm layer from June to September. *Liu et al. (2021)* discovered that the absolute change rate value of soil moisture is positively correlated with air temperature, relative air humidity, and rainfall, but negatively correlated with photosynthetically active radiation, vapor pressure deficits, and wind speed in different subtropical plantations in the Yangtze River Delta region. The response characteristics of soil moisture to meteorological factors are not consistent across different regions because of differences in climate, topography, soil texture, and vegetation. The dominant meteorological drivers for soil moisture variation across different periods are also not consistent. Precipitation, solar radiation, and air temperature are the main factors that affect soil moisture variability in water-scarce areas (*Liu et al., 2012a*; *Wang et al., 2019*; *Han et al., 2020*). Furthermore, air humidity, wind speed, and other factors influence changes in soil moisture by affecting the intensity of soil moisture evaporation (*Akinyemi et al., 2007*; *MacDonald, Pomeroy & Essery, 2018*). Soil temperature is another environmental factor that influences soil moisture movement; according to *Liu, Xia & Shang (2020)*, it directly impacts soil water movement and distribution, and is one of the major influencing factors affecting bare soil evaporation. Soil temperature profile, particularly that of surface soil, varies with seasons and day–night changes in nature and has a direct impact on soil moisture infiltration, redistribution, and evaporation, which is especially significant in arid and semi-arid regions (*Sarkar, Paramanick & Goswami, 2007*; *Sypka, Kucza & Starzak, 2016*).

Energy exchange between soil and the atmosphere causes changes in soil temperature owing to the combined effects of solar radiation and precipitation (*Meng et al., 2020*). Many academics have conducted extensive research on the relationship between changes in soil temperature and meteorological factors and have obtained important findings. According to *Hu & Feng (2005)*, air temperature has a large impact on soil temperature throughout
Eurasia across seasons, and precipitation influences soil temperature, particularly at high latitudes and during winter. *Sattari et al. (2020)* used tree-based hybrid data mining models to estimate soil temperature in Turkey's Sivas Divrigi district, and concluded that sunshine duration and air temperature are key factors in predicting soil temperature, whereas precipitation is the least important meteorological variable. *Dodds et al. (2003)* investigated the factors influencing soil temperature in pepper fields under plastic mulches, and discovered that mean air temperature and mean radiation, followed by wind speed and relative humidity, were the best predictors of soil temperature, whereas rainfall had little or no effect on it. It can be concluded that differences in the geographic environments and ecological factors of study areas can lead to differences in the effects of the same meteorological factors on soil temperature. Soil temperature is influenced not only by meteorological factors, but also by soil moisture. Moisture and heat in the soil influence and interact with one another, and changes in soil moisture can change the thermal characteristics of the soil, thereby affecting its temperature (*Cheng et al., 2013*). *Mi et al. (2014)* discovered that if soil bulk density remains constant during frequent dry–wet alternations, soil moisture has the largest influence on soil thermal parameters, and that the water retention effect of mulching has a direct impact on the dynamic changes in surface soil thermal parameters. In a potato field in Wuchuan County, Inner Mongolia, China, *Zhang et al. (2020)* discovered an inversely proportional relationship between soil moisture and temperature across treatments which had various water levels.

Soil moisture and temperature respond differently to environmental factors depending on the land use type in each region. For instance, according to *Hao et al. (2019)*, converting the vegetation of an area from grasses to evergreen woody plants prolongs the impact of meteorological drought on soil moisture. Therefore, restoring prairies that have been heavily encroached on by woody species may mitigate the impact of climate change on water resources in the transition zone of the United States. *Chen et al. (2022)* studied the response of soil moisture to precipitation under three typical land-use patterns in the Maowusu Desert of Ningxia, China, and found that the wet front of sand had the fastest response and the largest infiltration depth when compared with a grassland and shrub land. In the grassland, the soil water content fluctuated greatly and the cumulative infiltration amount was large, whereas, in the shrubland, the response of soil water content to rainfall events was relatively stable, and the infiltration amount was the lowest among the three (*Chen et al., 2022*). *Chen et al. (2009a)* found that although the soil temperature in winter wheat fields in the North China Plain under different tillage methods (plowing, rotary tillage, and no-tillage with straw mulching) significantly responded to air temperature changes, indicating that the tillage methods affected the range of change of soil temperature. *Sun et al. (2016)* studied the change in soil temperature and its response to different air temperatures under various irrigation modes in the Inner Mongolia Hetao Irrigation District and found that the temperature of the soil where drip irrigation was practiced under plastic film responded more quickly to changes in air temperature than that where surface irrigation was practiced. They also found that the soil temperature where drip irrigation was practiced under plastic film was higher than that where surface irrigation was practiced at the seedling stage of maize. Under the specific bio-climatic conditions in the loess hilly region, complex

and changeable landform types, deep loess, and various types of soil have developed from it. In particular, with the implementation of a project to revert farmland to forestland and grassland, a variety of vegetation types and land use patterns have been developed in the region, such as large areas of arable land mainly planted with maize, potatoes, soybeans, *etc*.; grasslands from farmlands conversion; and scattered woodlands (jujube orchards, *Caragana microphylla* woodlands, *Robinia pseudoacacia* woodlands, *etc*.), all of which have had considerable effects on soil moisture and temperature in the region. Currently, there are few studies on how soil moisture and temperature respond to environmental factors under various land use types, and the response of soil moisture and temperature to environmental factors needs to be clarified.

Therefore, our study investigated four typical land use types in the loess hilly region (soybean sloping field, maize terraced field, jujube orchard, and grassland). Pearson correlation, path, stepwise regression, and grey relational analyses were used to study the relationship between changes in soil moisture and temperature and major meteorological factors (temperature, humidity, solar radiation, wind speed, and precipitation) under different levels of annual precipitation for the four land use types, as well as the relationship between soil moisture and temperature. The goal of this research is to uncover the mechanisms by which various environmental factors influence soil moisture and temperature under different land use types to provide a scientific foundation to improve land use and enable vegetation restoration in the loess hilly region.

## MATERIALS & METHODS

### Overview of the study area

The study area was located in the Yuanzegou watershed (37°14′N, 118°21′E), Qingjian County (Fig. 1), Shaanxi Province, China, in the north-central part of the Loess Plateau, which is a typical loess hilly and gully area. The study area has a temperate continental monsoon climate, with an annual average air temperature of 8.6 °C. January experiences the lowest monthly average air temperature of −6.5 °C, while July has the highest monthly average air temperature of 22.8 °C. The average annual precipitation is 505 mm, but it is unevenly distributed throughout the year, with 70% of the precipitation occurring between July and September. The soil in the study area is loessial (which is comprised of largely silt, followed by sand, and then clay) and has a high infiltration capacity. The field capacity and wilting moisture content are approximately 25% and 7% (volumetric moisture content), respectively. The precipitation in the growing season (from May to October) in 2014 and 2015 was 377.4 mm and 289.2 mm (Fig. 2), respectively, and 2014 and 2015 were considered a normal precipitation year and a dry year, respectively, according to *Hao, Wei & Dang (2003)*.

### Experimental scheme

Considering the above-mentioned land use types and vegetation characteristics in the study area, soybean sloping fields, maize terraced fields, jujube orchards, and grasslands with similar slope aspects (shady slopes) and gradients (approximately 18°) were selected as the experimental plots (Fig. 3). Soybean and maize were sown at densities of $19.5 \times 10^4$

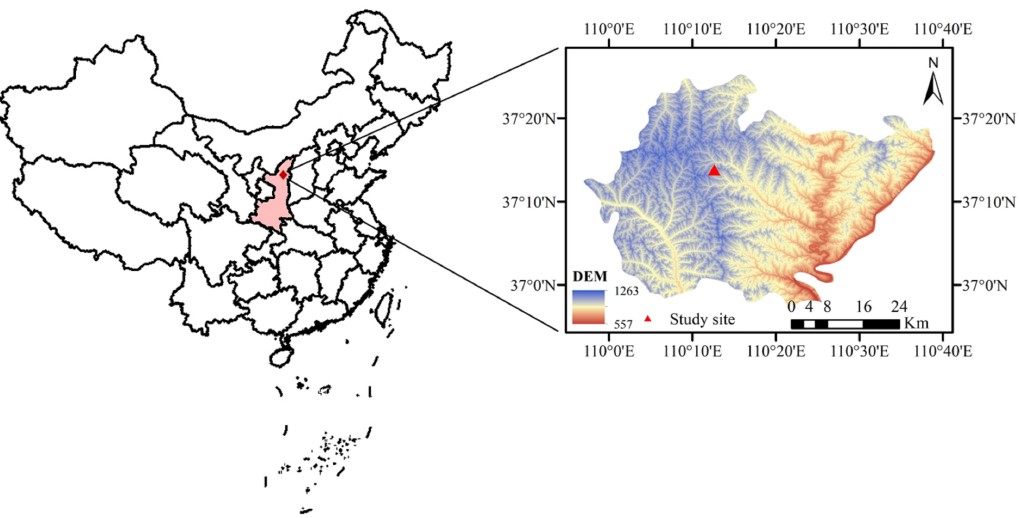

**Figure 1**   **Geographical location of the study area.**

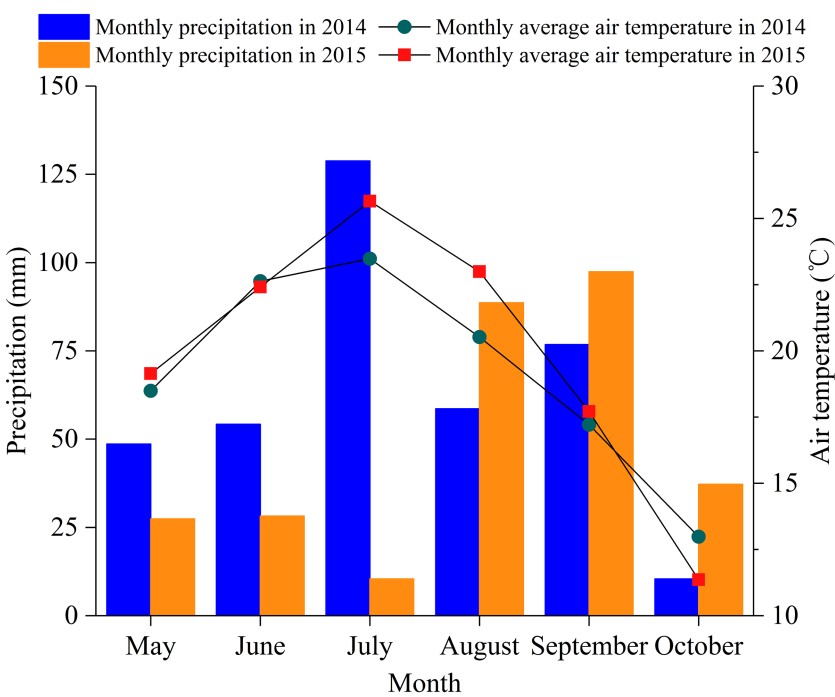

**Figure 2**   **Monthly precipitation and monthly mean air temperature during the 2014 and 2015 growing seasons in the study area.**

and $9 \times 10^4$ plants $hm^{-2}$, respectively, in late April and early May each year, and both were harvested in early October. Lizao was the jujube species used in the experiment. It had been planted in 2003 and was in the full-bearing period during the experiment. Plant spacing was 2 m, and row spacing was 3 m. The grassland has been naturally restored from

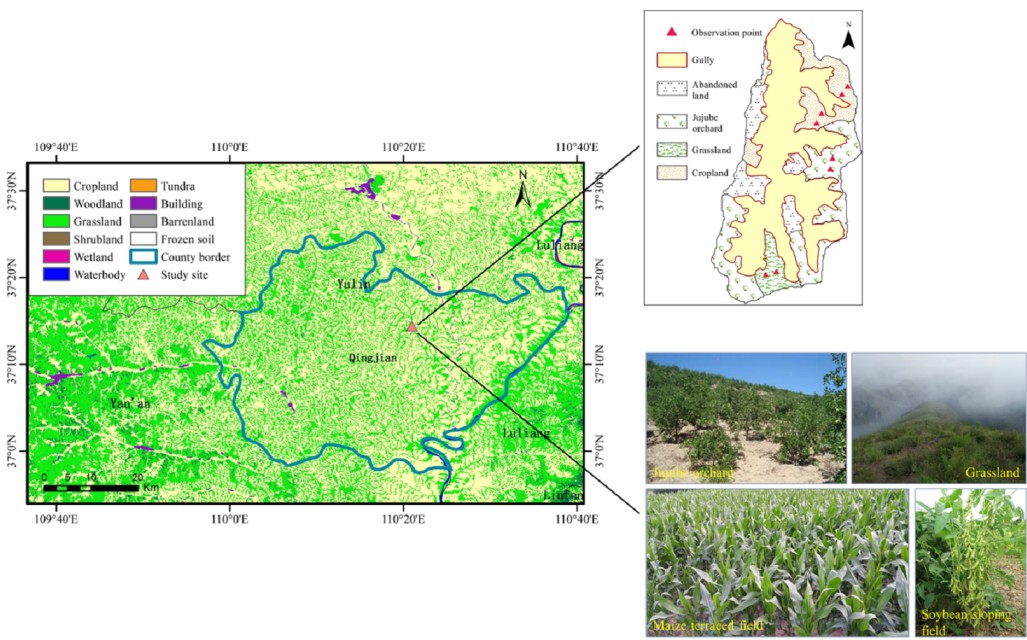

**Figure 3** **Distribution of land use types and experimental plots in the study area.**

sloping farmland for more than 30 years ago. The eugenic plant was *Artemisia gmelinii* and the associated plants were *Lespedeza daurica* (Laxm.) Schindl. and *Bothriochloa ischaemum* (L.) Keng. During the experiment, none of the plots were irrigated, and field management followed the local standard. The soil properties of the different sloping land use types are listed in Table 1.

Data were collected as previously described in *Zhang et al. (2022)*. Specifically, two sets of automatic soil moisture and temperature monitoring instruments were installed in the middle of each experimental plot along the same contour line with a 10-meter interval in April 2014. The monitoring points in the soybean sloping field and maize terraced field were placed between crop rows, and the jujube orchard monitoring points were placed 30 cm away from the trunks of the jujube trees. In this study, an EC-5 soil moisture sensor (Decagon Devices Inc., Pullman, WA, USA) was used to measure soil volumetric water content (VWC). Compared with traditional soil moisture sensors, such as the Hydra Probe and Theta Probe, EC-5 soil moisture sensors are economical dielectric sensors suitable for regional research. The soil volumetric water content can be calculated by quickly measuring the soil dielectric constant. It has a high-frequency signature that significantly reduces the influence of electroconductibility and soil texture on its readings and is suitable for all soil types. In addition, the high measurement frequency improves the sensor's accuracy ($\pm 2\%$ VWC) and range (0–100% VWC). It can realize fast, accurate, and continuous monitoring of soil moisture and is currently widely used in the real-time monitoring of soil moisture (*Deveci, Konukcu & Altürk, 2019*; *Lei et al., 2020*). The RR-7110 soil temperature sensor (Rainroot Scientific Ltd., Peking, CN) was used to measure soil temperature. Compared to traditional soil temperature sensors, the RR-7110 soil temperature sensor is easy to connect

**Table 1  Soil physical properties and nutrient content of experimental plots.**

| Sloping land use type | Soil layer/ cm | Soil texture | | | Bulk density/ (g cm⁻³) | Soil porosity/ % | Saturated hydraulic conductivity/ (cm d⁻¹) | Organic matter/ (g kg⁻¹) | Total N/ (g kg⁻¹) |
|---|---|---|---|---|---|---|---|---|---|
| | | Sand/% | Silt/% | Clay/% | | | | | |
| | 0–20 | 17.3 ± 2.8 | 63.6 ± 1.3 | 19.1 ± 3.4 | 1.28 ± 0.09 | 54.0 ± 3.4 | | 5.12 ± 0.79 | 0.34 ± 0.04 |
| Grassland | 20–40 | 15.1 ± 1.7 | 62.9 ± 1.6 | 22.0 ± 0.7 | 1.28 ± 0.04 | 52.1 ± 1.5 | 35.3 ± 8.3 | 4.43 ± 0.28 | 0.28 ± 0.04 |
| | 40–60 | 14.9 ± 3.7 | 63.3 ± 1.3 | 21.8 ± 3.8 | – | – | | 4.23 ± 0.36 | 0.28 ± 0.06 |
| | 0–20 | 21.0 ± 5.6 | 63.0 ± 3.7 | 16.0 ± 3.9 | 1.17 ± 0.15 | 58.9 ± 5.6 | | 3.34 ± 0.24 | 0.29 ± 0.02 |
| Soybean sloping field | 20–40 | 19.5 ± 5.4 | 63.4 ± 2.4 | 17.0 ± 4.5 | 1.29 ± 0.11 | 54.3 ± 3.7 | 74.2 ± 20.6 | 2.66 ± 0.31 | 0.22 ± 0.04 |
| | 40–60 | 19.9 ± 3.9 | 65.0 ± 1.9 | 15.1 ± 3.2 | – | – | | 2.52 ± 0.38 | 0.20 ± 0.02 |
| Maize terraced field | 0–20 | 17.7 ± 1.9 | 63.8 ± 2.3 | 18.5 ± 2.9 | 1.26 ± 0.11 | 55.1 ± 4.2 | | 4.24 ± 0.37 | 0.30 ± 0.03 |
| | 20–40 | 16.6 ± 3.9 | 64.9 ± 1.7 | 18.6 ± 4.1 | 1.36 ± 0.07 | 52.5 ± 2.7 | 55.6 ± 10.4 | 2.98 ± 0.33 | 0.22 ± 0.03 |
| | 40–60 | 16.2 ± 4.1 | 63.3 ± 1.1 | 20.5 ± 4.2 | – | – | | 2.74 ± 0.42 | 0.20 ± 0.03 |
| | 0–20 | 23.7 ± 4.0 | 62.6 ± 1.9 | 13.7 ± 2.4 | 1.31 ± 0.12 | 51.7 ± 4.5 | | 3.64 ± 0.85 | 0.31 ± 0.06 |
| Jujube orchard | 20–40 | 21.6 ± 3.4 | 64.0 ± 1.9 | 14.4 ± 2.7 | 1.41 ± 0.10 | 52.1 ± 3.8 | 36.6 ± 9.6 | 2.76 ± 0.89 | 0.26 ± 0.05 |
| | 40–60 | 20.7 ± 2.2 | 64.2 ± 1.2 | 15.1 ± 2.5 | – | – | | 2.60 ± 0.66 | 0.27 ± 0.05 |

**Notes.**
The sampling date is September 5, 2014. Data are mean ± SD, $n = 3$. Soil particle composition: Sand% (0.02–2 mm), Silt% (0.002–0.02 mm), and Clay% (<0.002 mm).

and install. It is suitable for continuous monitoring of both wild and harsh environments. It has low power consumption and has the function of automatic data saving when powered off, and also has high measurement accuracy (±0.2 °C) and resolution (0.02 °C); it is therefore widely used in soil heat monitoring in agriculture, forestry, ecology and other fields (*Duan, Yang & Mao, 2018*; *Ma et al., 2021*). The probes for the soil moisture sensor were placed at depths of 10, 20, 60, 100, and 160 cm. The probes for the soil temperature sensor were placed at depths of 10, 20, 40, 60, and 100 cm. During the vegetation growing season (May to October) in 2014 and 2015, soil moisture and temperature were measured every 2 min, and the data were recorded every 10 min. To characterize the soil moisture and temperature at a certain depth under each sloping land use type, we averaged the soil moisture and temperature at the same depth at the two monitoring points under this sloping land use type. An AR5 automatic weather station (Avolon Scientific Inc., Jersey City, NJ, USA) continuously monitored weather variables, such as temperature, humidity, pressure, solar radiation, wind speed, and precipitation in the study area.

## Data processing
### Path analysis
Statistical path analysis was used to decompose the correlation coefficients. Its importance lies not only in revealing the direct and indirect influence of $x_i$ on $y$ in correlation analysis of multiple independent variables $x_1, x_2, \ldots, x_m, y$, but also in obtaining the path information of the best influence on $y$ from the relationship between an independent variable and other independent variables in a complex correlation between $x_1, x_2, \ldots, x_m, y$. Therefore, the absolute value of the path coefficient can be used to directly compare the key role of each independent variable in the regression equation, which is of great practical value for
clarifying key factors and changing the reflection of dependent variables in a multivariable system. Overall, path analysis is more comprehensive and delicate than correlation analysis. SPSS linear regression was used to perform the path analysis in this study. Please refer to *Du & Chen (2010)* for more information on the specific procedures. It is worth noting that this study used stepwise regression to create a linear regression equation and then calculated the path coefficient. Stepwise regression has the advantage of gradually adding or removing an independent variable from all available independent variables until the best regression equation is obtained.

### Grey relational analysis

The basic principle of grey relational analysis (GRA) is to compare the geometric relationships of statistical sequences to determine the closeness of multiple factors in the system (*i.e.,* grey relational grade). The greater the grey relational grade, the closer the geometric shape of the sequence curves, and vice versa. GRA has an advantage over traditional statistical analysis or other analysis methods in that it analyzes factors based on their development trends, does not specify the sample size, does not require a typical distribution law, has a small calculation amount, and provides calculation results that are consistent with qualitative analysis results.

Before performing GRA, the reference sequence must first be determined, followed by a comparison of the similarity of the other sequences to the reference sequence. If the reference sequence is $X_0 = \{X_0(k)|k = 1, 2, \ldots, n\}$, and the comparison sequence is $X_i = \{X_i(k)|k = 1, 2, \ldots, n\}(i = 1, 2, \ldots, m)$, then the correlation coefficients of $X_i$ $(k)$ and $X_0$ $(k)$ are calculated as follows:

$$\varepsilon_i(k) = \frac{\min_i \min_k |x_0(k) - x_i(k)| + \rho \max_i \max_k |x_0(k) - x_i(k)|}{|x_0(k) - x_i(k)| + \rho \max_i \max_k |x_0(k) - x_i(k)|} \tag{1}$$

where $\rho$ is the resolution coefficient and is usually 0.5; $|x_0(k) - x_i(k)|$ is the absolute difference between $X_0$ and the $k$-th index of $X_i$; $\min_i \min_k |x_0(k) - x_i(k)|$ and $\max_i \max_k |x_0(k) - x_i(k)|$ are the two-level minimum and maximum differences, respectively.

The following formula can be used to calculate the correlation coefficient between $X_i$ $(k)$ and $X_0$ $(k)$.

$$\varepsilon_i(k) = \{\varepsilon_i(k)|k = 1, 2, \ldots, n\}. \tag{2}$$

The correlation coefficient value of each comparison and reference sequence at each point is obtained from the correlation coefficient calculation. There are numerous outcomes and the data are dispersed. As a result, the correlation coefficient of each comparison sequence at each point must be collectively reflected in one value, which is the grey relational grade $r$ $(x_0, x_i)$ of the comparison sequence to the reference sequence, commonly abbreviated as $r_i$. The average method is the most commonly used method for calculating the grey relational grade, and the formula is as follows:

$$\gamma_i = \frac{1}{n} \sum_{k=1}^{n} \varepsilon_i(k). \tag{3}$$

The grey relational grade is as previously described by *Zhang et al. (2021)*. Specifically, the grey relational grade in this study reflects the proximity of each influencing factor to soil temperature. The greater the grey relational grade, and the closer it is to 1, the closer the connection between the reference sequence and the comparison sequence, and the greater the influence of the comparison sequence on the reference sequence, and vice versa. When the grey relational grade is greater than 0.80, the factors corresponding to the comparison sequence are thought to be closely related to soil temperature and have a large impact. The soil temperatures of the surface layer (0–20 cm), middle layer (20–60 cm), and deep layer (60–100 cm) in the four experimental sloping land use types were used as parameter variables. To determine the effects of air temperature, air humidity, solar radiation, wind speed, precipitation, and soil moisture on soil temperature under various sloping land use types, the reference sequence and each environmental factor sequence were quantified using dimensionless equations, and the grey relational coefficients between each environmental factor sequence and the reference sequence were calculated.

## Data analysis

Binary correlation analysis between soil moisture and environmental factors was performed using the Pearson correlation analysis method in SPSS software (SPSS Inc., Chicago, Illinois, USA). The stepwise regression method in SPSS was used to screen the independent variables in the stepwise regression analysis of the environmental factors affecting soil moisture, and the variables that met the allowable level (0.05) were entered into the model. Drawing was performed using OriginPro 2017 software (OriginLab Corp., Northampton, MA, USA).

# RESULTS

## Soil moisture response to environmental factors
### Binary correlation analysis between soil moisture and environmental factors

According to previous research (*Tang et al., 2019a*), the soil moisture in the 0–60 cm soil layer fluctuated greatly during the growing season, and the soil moisture in the 60–160 cm soil layer altered uniformly under various experimental sloping land use types. It can be concluded that environmental factors had a significant impact on soil moisture in the 0–60 cm soil layer. Table 2 shows the correlation analysis results for daily mean moisture content in the 0–60 cm soil layer and daily mean air temperature ($x_1$), daily mean air humidity ($x_2$), daily mean solar radiation ($x_3$), daily mean wind speed ($x_4$), daily precipitation ($x_5$), and daily mean soil temperature in the 0–60 cm soil layer ($x_6$) under the four experimental sloping land use types.

During the 2014 growing season, soil moisture in the 0–60 cm soil layer of the soybean sloping field was significantly correlated with air temperature, air humidity, and wind speed, with the correlation between soil moisture, air temperature, and air humidity reaching a significant level ($p < 0.01$) (Table 2). In the 2015 growing season, the soil moisture of the soybean sloping field was significantly correlated with the air temperature, air humidity, and solar radiation ($p < 0.01$). It can be concluded that, whether in a normal precipitation year or a dry year, air temperature and humidity were the main controlling

**Table 2** Correlation analysis between soil moisture in 0–60 cm soil layer and main environmental factors under experimental sloping land use types.

| Year | Sloping land use type | Environmental factor | | | | | |
|------|-----------------------|-----|-----|-----|-----|-----|-----|
| | | $x_1$ | $x_2$ | $x_3$ | $x_4$ | $x_5$ | $x_6$ |
| 2014 | Grassland | −0.422** | 0.511** | −0.151 | −0.222** | 0.217** | −0.322** |
| | Soybean sloping field | −0.294** | 0.324** | −0.031 | −0.174* | 0.15 | −0.305** |
| | Maize terraced field | −0.242** | 0.402** | −0.035 | −0.265** | 0.145 | −0.224** |
| | Jujube orchard | −0.289** | 0.361** | −0.003 | −0.157 | 0.172* | −0.378** |
| 2015 | Grassland | −0.698** | 0.227** | −0.334** | −0.111 | 0.003 | −0.758** |
| | Soybean sloping field | −0.747** | 0.246** | −0.384** | −0.058 | 0.025 | −0.779** |
| | Maize terraced field | −0.152 | 0.282** | −0.131 | 0.375** | 0.198** | −0.431** |
| | Jujube orchard | −0.411** | 0.027 | −0.075 | 0.245** | 0.025 | −0.520** |

**Notes.**

$x_1$, daily mean air temperature; $x_2$, daily mean air humidity; $x_3$, daily mean solar radiation; $x_4$, daily mean wind speed; $x_5$, daily precipitation; $x_6$, daily mean soil temperature in the 0–60 cm soil layer.

*Significance at 0.05 level.

**Significance at 0.01 level; similarly hereinafter.

factors for soil moisture changes in the soybean sloping field at a small catchment scale. The soil moisture of the maize terraced field was significantly correlated with air temperature, humidity, and wind speed ($p < 0.01$) during the 2014 growing season. The correlations between soil moisture in the maize terraced field and air humidity, wind speed, and precipitation were significant ($p < 0.01$) during the 2015 growing season. Based on the findings, air humidity and wind speed were the most important controlling factors affecting soil moisture changes in the maize terraced field on a regional scale during various levels of annual precipitation. In the 2014 growing season, soil moisture of the jujube orchard was significantly correlated with air temperature, air humidity, and precipitation; however, in the 2015 growing season, it was only significantly correlated with air temperature and wind speed ($p < 0.01$). In both growing seasons, the correlation between soil moisture in the jujube orchard and air temperature was significant ($p < 0.01$), indicating that among the meteorological factors studied, air temperature had a significant impact on soil moisture in the jujube orchard (*i.e.,* soil moisture decreased as air temperature rose). During the 2014 and 2015 growing seasons, the correlation between soil moisture, air temperature, and air humidity in the grassland reached a significant level ($p < 0.01$), indicating that air temperature and air humidity were the main factors affecting the change in soil moisture in the grassland, regardless of precipitation differences. In the growing seasons of 2014 and 2015, air temperature, solar radiation, and wind speed were negatively correlated with soil moisture under the four sloping land use types. This was due to increased air temperature, solar radiation, and wind speed, which resulted in increased soil evaporation, which subsequently resulted in a decrease in the moisture content in the 0–60 cm soil layer. Under various sloping land use types, air humidity and precipitation were positively correlated with soil moisture, indicating that as air humidity and precipitation increased, soil moisture also increased.

In the 2014 and 2015 growing seasons, there was a negative correlation between soil moisture and soil temperature in the 0–60 cm soil layer under various sloping land use

types. All of them were found to be statistically significance ($p < 0.01$) (Table 2), indicating that soil moisture and soil temperature affected each other in this soil layer, and as the soil temperature increased, the soil moisture decreased. The dry year showed a substantially higher correlation between soil moisture and soil temperature in the 0–60 cm soil layer under the same sloping land use type as compared to that in the normal precipitation year. This could be because the dry year experienced a severe soil water shortage, and the high soil temperature in the 0–60 cm soil layer increased soil moisture evaporation, resulting in a strong negative correlation between soil temperature and soil moisture. In the 2014 and 2015 growing seasons, a lower correlation between soil moisture and soil temperature in the 0–60 cm soil layer was found in the maize terraced field than in the other three sloping land use types. This is because the soil temperature of the 0–60 cm soil layer in the maize terraced field was relatively low, and the average soil temperature of this soil layer did not exceed 20 °C in the two growing seasons (*i.e.,* soil moisture evaporation was minimal). Similarly, the average soil temperature of the 0–60 cm soil layer in the jujube orchard was 20.73 °C in the 2014 growing season, which was 0.41, 1.20, and 1.81 °C higher than that of the soybean sloping field, maize terraced field, and grassland, respectively. The evaporation of soil moisture in this soil layer was accelerated by the higher soil temperature in the jujube orchard, causing the soil temperature to have a strong negative effect on soil moisture.

### Path analysis of the relationship between soil moisture and environmental variables

It is easy to overlook the interaction between environmental factors only by judging their contribution of environmental factors to soil moisture based on the simple correlation coefficient between environmental factors and soil moisture. The correlation coefficients between various environmental factors and soil moisture were divided into direct and indirect effects for path analysis, to further explore the direct and indirect effects of various environmental factors on soil moisture.

In the 2014 growing season, air humidity had the greatest direct effect on soil moisture in the 0–60 cm soil layer of the soybean sloping field, maize terraced field, and grassland, with direct path coefficients of 0.492, 0.491, and 0.716, respectively, followed by solar radiation (Table 3). Soil temperature had the greatest direct effect on soil moisture in the jujube orchard, with a direct path coefficient of −1.101. The direct influence of air humidity and solar radiation on soil moisture under the four sloping land use types showed positive effects. The direct impact of soil temperature on soil moisture in soybean sloping fields, jujube orchards, and grasslands was negative. The absolute value of the direct path coefficient of soil temperature in the soybean sloping field and grassland was greater than the absolute value of the sum of indirect path coefficients. The sum of indirect effect coefficients of soil temperature was very small, indicating that the influence of soil temperature on the soil moisture of the soybean sloping field and grassland in the 0–60 cm soil layer was mainly reflected in the direct effect. The evaporation of soil moisture in the soybean sloping field and grassland intensified as the soil temperature increased, reducing the moisture content in the 0–60 cm soil layer. As a result, the impact of soil temperature on soil moisture was primarily a direct effect, with little correlation to other meteorological
**Table 3 Path analysis of influencing factors on soil moisture in 0–60 cm soil layer under different sloping land use types in the 2014 growing season.**

| Sloping land use type | Environmental factor | Simple correlation coefficient with $y$ | Direct path coefficient | Indirect path coefficient | | | | | |
|---|---|---|---|---|---|---|---|---|---|
| | | | | $x_1$ | $x_2$ | $x_3$ | $x_4$ | $x_6$ | Total |
| Grassland | $x_2$ | 0.511 | 0.716 | | | −0.603 | | −0.139 | −0.205 |
| | $x_3$ | −0.151 | 0.429 | | −0.603 | | | 0.384 | −0.581 |
| | $x_6$ | −0.322 | −0.388 | | −0.139 | 0.384 | | | 0.065 |
| Soybean sloping field | $x_2$ | 0.324 | 0.492 | | | −0.593 | | −0.207 | −0.167 |
| | $x_3$ | −0.031 | 0.411 | | −0.593 | | | 0.406 | −0.442 |
| | $x_6$ | −0.305 | −0.370 | | −0.207 | 0.406 | | | 0.065 |
| Maize terraced field | $x_1$ | −0.242 | −0.266 | | −0.476 | 0.591 | −0.014 | | 0.024 |
| | $x_2$ | 0.402 | 0.491 | −0.476 | | −0.603 | −0.249 | | −0.089 |
| | $x_3$ | −0.035 | 0.432 | 0.591 | −0.603 | | 0.075 | | −0.467 |
| | $x_4$ | −0.265 | −0.179 | −0.014 | −0.249 | 0.075 | | | −0.086 |
| Jujube orchard | $x_1$ | −0.289 | 0.801 | | −0.477 | 0.597 | | 0.899 | −1.090 |
| | $x_2$ | 0.361 | 0.742 | −0.477 | | −0.605 | | −0.234 | −0.382 |
| | $x_3$ | −0.003 | 0.425 | 0.597 | −0.605 | | | 0.415 | −0.428 |
| | $x_6$ | −0.378 | −1.101 | 0.899 | −0.234 | 0.415 | | | 0.723 |

factors. The absolute values of the direct path coefficients of air temperature, air humidity, and wind speed were greater than the absolute value of the sum of their respective indirect path coefficients in the maize terraced field. The sum of these three meteorological factors' respective indirect path coefficients was small, indicating that the influence of these three meteorological factors on the soil moisture of the 0–60 cm soil layer of the maize terraced field was mainly reflected in the direct effect. The evaporation capacity of the atmosphere increased as the air temperature increased, and the increase in wind speed aided the increase in the evaporation rate. Because of the aforementioned comprehensive factors, soil evaporation increased in the maize terraced field, reducing the moisture content of the 0–60 cm soil layer. As a result, the influence of air temperature and wind speed on soil moisture was mainly a direct effect, and the correlation with other environmental factors was weak. Soil evaporation decreased as air humidity increased and the rate of soil moisture loss slowed. It has a direct influence on soil moisture, with little correlation with other influencing factors. Although solar radiation had a large direct effect on soil moisture under the four sloping land use types, all of which had coefficients above 0.4, it also had a relatively large indirect effect on soil moisture through other environmental factors (such as air temperature, air humidity, and soil temperature), resulting in a small overall impact of solar radiation on soil moisture. Therefore, the simple correlation coefficients between solar radiation and soil moisture under various sloping land use types were low.

Soil temperature had the greatest direct impact on soil moisture in the 0–60 cm soil layer under the soybean sloping field, maize terraced field, jujube orchard, and grassland in the 2015 growing season, with direct path coefficients of −0.762, −0.861, −0.950,

Table 4 Path analysis of influencing factors on soil moisture in 0–60 cm soil layer under different sloping land use types in the 2015 growing season.

| Sloping land use type | Environmental factor | Simple correlation coefficient with $y$ | Direct path coefficient | Indirect path coefficient | | | | |
|---|---|---|---|---|---|---|---|---|
| | | | | $x_1$ | $x_2$ | $x_4$ | $x_6$ | Total |
| Grassland | $x_2$ | 0.227 | 0.130 | | | | −0.131 | 0.097 |
| | $x_6$ | −0.758 | −0.741 | | −0.131 | | | −0.017 |
| Soybean sloping field | $x_2$ | 0.246 | 0.173 | | | | −0.096 | 0.073 |
| | $x_6$ | −0.779 | −0.762 | | −0.096 | | | −0.017 |
| Maize terraced field | $x_1$ | −0.152 | 0.576 | | | −0.064 | 0.830 | −0.729 |
| | $x_4$ | 0.375 | 0.225 | −0.064 | | | −0.217 | 0.150 |
| | $x_6$ | −0.431 | −0.861 | 0.830 | | −0.217 | | 0.429 |
| Jujube orchard | $x_1$ | −0.411 | 0.549 | | −0.403 | −0.065 | 0.913 | −0.960 |
| | $x_2$ | 0.027 | 0.193 | −0.403 | | −0.505 | −0.177 | −0.166 |
| | $x_4$ | 0.245 | 0.223 | −0.065 | −0.505 | | −0.163 | 0.022 |
| | $x_6$ | −0.520 | −0.950 | 0.913 | −0.177 | −0.163 | | 0.431 |

and −0.741, respectively, all of which had negative effects (Table 4). The soil moisture of the soybean sloping field, jujube orchard, and grassland was positively affected by air humidity. The direct path coefficients of soil temperature under soybean sloping fields and grasslands were greater than the sum of indirect path coefficients, and the sum of indirect path coefficients of soil temperature was very low (−0.017), indicating that the influence of soil temperature on soil moisture under these two sloping land use types was mainly reflected in the direct effect, with little correlation with other meteorological factors. The evaporation of soil moisture was accelerated by an increase in soil temperature, resulting in soil moisture loss and decline. The direct effect (0.193) of air humidity on soil moisture in jujube orchards was opposite to the comprehensive indirect effect (−0.166) on soil moisture through the influence of air temperature, wind speed, and soil temperature. The simple correlation coefficient between air humidity and soil moisture in jujube orchards was as low as 0.027 owing to the superposition of the two effects, indicating that atmospheric humidity had negligible effect on the change in soil moisture in jujube orchards and that it was unnecessary to consider too much. The direct effect of air temperature on soil moisture in the maize terraced field and jujube orchard was positive; however, the indirect effect on soil moisture by influencing wind speed and soil temperature was negative, and the indirect path coefficient was approximately 1.5 times that of the direct path coefficient. The comprehensive effect of air temperature on soil moisture in the maize terraced field and jujube orchard was negative, as predicted by its indirect effect.

### Environmental factors affecting soil moisture: a stepwise regression analysis

The effects of different independent variables on the dependent variable can be explained using multiple stepwise regression analyses. Therefore, the dependent variable in this study was soil moisture ($y$) in the 0–60 cm soil layer under various sloping land use types, and

the independent variables were air temperature ($x_1$), air humidity ($x_2$), solar radiation ($x_3$), wind speed ($x_4$), precipitation ($x_5$), and soil temperature ($x_6$) in the 0–60 cm soil layer. The main environmental factors affecting soil moisture were investigated using multiple stepwise regression analysis. Empirical models were developed based on the final regression results to predict the soil moisture content in the 0–60 cm soil layer under different sloping land use types (Table 5).

The regression equations for various sloping land use types reached extremely significant levels ($p < 0.01$) during the 2014 and 2015 growing seasons (Table 5). Different environmental factors were entered into the stepwise regression model for different sloping land use types in different precipitation years. In the 2014 growing season, the contribution rates of air humidity and solar radiation to soil moisture under the soybean sloping field, maize terraced field, and grassland were higher than those of other environmental factors; the total contribution rates of air humidity and solar radiation to soil moisture were 70.91% for soybean sloping fields, 67.48% for maize terraced fields, and 74.68% for grasslands. The results showed that air humidity and solar radiation were the main controlling factors affecting soil moisture in the soybean sloping field, maize terraced field, and grassland during the 2014 growing season. In the jujube orchard, soil temperature contributed more to soil moisture (35.87%) than air temperature, air humidity, or solar radiation, indicating that the relationship between soil temperature and soil moisture in the 0–60 cm soil layer was closer in the 2014 growing season. During the 2015 growing season, soil temperature was entered into the regression equation of soil moisture in the 0–60 cm soil layer under four sloping land use types, and the contribution rate of soil temperature to soil moisture under all sloping land use types was more than 49%, with more than 81% for soybean sloping fields and grasslands. Soil temperature had a significant effect on soil moisture change under various land use types in the dry year, and it was the main controlling factor of soil moisture change in soybean sloping fields and grasslands, according to the findings. The sum of air temperature and soil temperature contributions to soil moisture in the maize terraced field and jujube orchard was 86.45% and 78.29%, respectively, indicating that air temperature and soil temperature were the main factors affecting soil moisture changes in the 0–60 cm soil layer of these two sloping land use types.

## Response of soil temperature to changes in environmental factors

The GRA was used in this study to examine the impact of different environmental factors on soil temperature, with soil temperature as the dependent variable and air temperature, air humidity, solar radiation, wind speed, precipitation, and soil moisture as the independent variables. The main factors influencing soil temperature were then screened for four different sloping land use types.

### GRA on the main influencing factors of soil temperature

The grey relational grade between environmental factors and soil temperature is shown in Table 6 for different sloping land use types. The grey relational grade between air temperature and soil temperature in the surface and middle layers under the four sloping land use types was the highest, ranging from 0.8275 to 0.8446, according to the standard

Tang et al. (2022), *PeerJ*, DOI 10.7717/peerj.13736

**Table 5  Stepwise regression analysis of environmental factors affecting soil moisture.**

| Year | Sloping land use type | Multiple regression equation | n | F | p | Total variance explained/% |
|---|---|---|---|---|---|---|
| 2014 | Grassland | $y = 11.997 + 0.086x_2 + 0.012x_3 - 0.234x_6$ | 157 | 37.834 | 0.000** | $x_2$ (46.68), $x_3$ (28.00), $x_6$ (25.32) |
| | Soybean sloping field | $y = 13.187 + 0.059x_2 + 0.011x_3 - 0.256x_6$ | 156 | 17.774 | 0.000** | $x_2$ (38.61), $x_3$ (32.30), $x_6$ (29.09) |
| | Maize terraced field | $y = 14.117 - 0.069x_1 + 0.033x_2 + 0.007x_3 - 0.629x_4$ | 157 | 15.823 | 0.000** | $x_1$ (19.45), $x_2$ (35.92), $x_3$ (31.56), $x_4$ (13.07) |
| | Jujube orchard | $y = 9.989 + 0.392x_1 + 0.095x_2 + 0.012x_3 - 0.724x_6$ | 150 | 29.740 | 0.001** | $x_1$ (26.09), $x_2$ (24.18), $x_3$ (13.86), $x_6$ (35.87) |
| 2015 | Grassland | $y = 17.222 + 0.014x_2 - 0.393x_6$ | 184 | 130.999 | 0.000** | $x_2$ (14.89), $x_6$ (85.11) |
| | Soybean sloping field | $y = 17.946 + 0.019x_2 - 0.469x_6$ | 184 | 158.103 | 0.000** | $x_2$ (18.49), $x_6$ (81.51) |
| | Maize terraced field | $y = 14.698 + 0.119x_1 + 0.550x_4 - 0.268x_6$ | 184 | 34.696 | 0.000** | $x_1$ (34.67), $x_4$ (13.56), $x_6$ (51.78) |
| | Jujube orchard | $y = 10.964 + 0.156x_1 + 0.014x_2 + 0.749x_4 - 0.361x_6$ | 184 | 21.896 | 0.000** | $x_1$ (28.66), $x_2$ (10.06), $x_4$ (11.65), $x_6$ (49.63) |

grey relational grade being greater than or equal to 0.80, indicating that air temperature was the primary factor affecting soil temperature in the 0–60 cm soil layer under various sloping land use types. The upper soil layer increased the barrier between the lower soil layer and the environment as the soil depth increased, and the grey relational grade between the soil temperature and air temperature decreased. Using the 2014 growing season as an example, the grey relational grade between the soil temperature in the surface layer and air temperature in the soybean sloping field, maize terraced field, jujube orchard, and grassland was 0.8439, 0.8444, 0.8398, and 0.8417, respectively, whereas the grey relational grade in the middle layer decreased to 0.8394, 0.8388, 0.8318, respectively, and 0.8395, indicating that the grey relational grade decreased with the deepening of the soil layer. Furthermore, solar radiation had a significant impact on the soil temperature in the surface layer under the soybean sloping field and jujube orchard, with grey relational grades of 0.8107 and 0.8006, respectively. During the 2014 growing season, the grey relational grade between soil temperature in the deep layer and air humidity was at its highest, with values of 0.8344, 0.8352, and 0.8356, under the soybean sloping field, maize terraced field, and grassland, respectively. Wind speed was closely related to soil temperature in the deep layer of the soybean sloping field, maize terraced field, and grassland in the 2015 growing season, with the grey relational grade exceeding 0.82, indicating that wind speed had a significant impact on soil temperature in the 60–100 cm soil layer under these three sloping land use types. Air temperature was the most crucial factor affecting soil temperature in the deep layer of the jujube orchard during the 2014 and 2015 growing seasons. The grey relational grade between precipitation and soil temperature in the surface layer and deep layer under the four sloping land use types was the lowest in the 2014 growing season, with the grey relational grade in the surface layer ranging from 0.5233 to 0.5605 and the grey relational grade in the deep layer not exceeding 0.6305, indicating that precipitation had insignificant effect on soil temperature in the 0–20 cm and 60–100 cm soil layers. The average grey relational grade between wind speed and soil temperature in the middle layer under the four sloping land use types was as low as 0.54, indicating that it was not closely related to the soil temperature in the 20–60 cm soil layer. The grey relational grades of precipitation, air humidity, solar radiation, and soil temperature in the surface, middle, and deep layers under various sloping land use types in the 2015 growing season were all low, with average grey relational grades of 0.6161, 0.5980, and 0.5856, respectively, indicating that precipitation, air humidity, and solar radiation had little effect on soil temperature in the 0–20 cm, 20–60 cm, and 60–100 cm soil layers.

There were differences in the grey relational grade between the soil temperature in the same soil layer and the same environmental factor under different sloping land use types and the same external meteorological conditions (Table 6), which may be caused by differences in the underlying surface of the four sloping land use types. When it comes to solar radiation, for example, the more vegetation on the underlying surface, the less solar radiation the ground receives, and the slower the soil's response to solar heating. Furthermore, the greater the surface roughness of the underlying surface, the lower the surface albedo and the easier it is to absorb solar radiation. It can be concluded that sloping land use influences the energy exchange between meteorological factors and soil, resulting

**Table 6  Grey relational grade between different environmental factors and soil temperature.**

| Year | Sloping land use type | Soil layer (cm) | Environmental factor | | | | | |
|---|---|---|---|---|---|---|---|---|
| | | | Air temperature | Air humidity | Solar radiation | Wind speed | Precipitation | Soil moisture |
| 2014 | Grassland | 0–20 | 0.8417 | 0.6916 | 0.7977 | 0.6218 | 0.5233 | 0.7327 |
| | | 20–60 | 0.8395 | 0.6342 | 0.7889 | 0.5349 | 0.6713 | 0.7081 |
| | | 60–100 | 0.6819 | 0.8356 | 0.6673 | 0.7302 | 0.5679 | 0.6826 |
| | Soybean sloping field | 0–20 | 0.8439 | 0.7150 | 0.8107 | 0.6249 | 0.5317 | 0.7450 |
| | | 20–60 | 0.8394 | 0.6479 | 0.7742 | 0.5482 | 0.6763 | 0.7040 |
| | | 60–100 | 0.6530 | 0.8344 | 0.6816 | 0.7258 | 0.6305 | 0.6725 |
| | Maize terraced field | 0–20 | 0.8444 | 0.6750 | 0.7952 | 0.6254 | 0.5342 | 0.7132 |
| | | 20–60 | 0.8388 | 0.7083 | 0.7847 | 0.5257 | 0.6742 | 0.6566 |
| | | 60–100 | 0.6670 | 0.8352 | 0.6857 | 0.6938 | 0.6102 | 0.7293 |
| | Jujube orchard | 0–20 | 0.8398 | 0.6864 | 0.8006 | 0.6098 | 0.5605 | 0.7433 |
| | | 20–60 | 0.8318 | 0.6888 | 0.6989 | 0.5522 | 0.6177 | 0.7884 |
| | | 60–100 | 0.8319 | 0.7644 | 0.7069 | 0.6787 | 0.5712 | 0.6419 |
| 2015 | Grassland | 0–20 | 0.8379 | 0.6796 | 0.6992 | 0.6639 | 0.6091 | 0.7271 |
| | | 20–60 | 0.8284 | 0.5574 | 0.6736 | 0.6849 | 0.6608 | 0.7000 |
| | | 60–100 | 0.7002 | 0.6768 | 0.5840 | 0.8235 | 0.5968 | 0.7015 |
| | Soybean sloping field | 0–20 | 0.8446 | 0.6879 | 0.7102 | 0.6757 | 0.6012 | 0.7411 |
| | | 20–60 | 0.8282 | 0.5711 | 0.6820 | 0.6853 | 0.6452 | 0.7003 |
| | | 60–100 | 0.6562 | 0.7032 | 0.5656 | 0.8268 | 0.5910 | 0.6973 |
| | Maize terraced field | 0–20 | 0.8439 | 0.6753 | 0.7202 | 0.6824 | 0.6060 | 0.7039 |
| | | 20–60 | 0.8301 | 0.6453 | 0.6836 | 0.6970 | 0.6711 | 0.7010 |
| | | 60–100 | 0.6977 | 0.6745 | 0.5834 | 0.8248 | 0.6542 | 0.7098 |
| | Jujube orchard | 0–20 | 0.8320 | 0.6973 | 0.7192 | 0.6780 | 0.6482 | 0.7389 |
| | | 20–60 | 0.8275 | 0.6180 | 0.7111 | 0.6841 | 0.6824 | 0.7125 |
| | | 60–100 | 0.8240 | 0.6716 | 0.6094 | 0.6776 | 0.6186 | 0.7022 |

in different responses of soil temperature to meteorological factors. The distribution and changes in soil moisture, as well as soil temperature, are affected by differences in the properties of the underlying surface under various sloping land use types.

### Soil temperature response characteristics to air temperature

According to the findings, air temperature had a significant impact on soil temperature under various experimental sloping land use types, particularly in the 0–60 cm soil layer. The correlation between soil temperature and air temperature at different depths was investigated using average data from daily observations, as shown in Fig. 4.

The coefficient of determination ($R^2$) between air temperature and soil temperature in the surface layer under the four sloping land use types ranged from 0.72 to 0.91 in the 2014 and 2015 growing seasons (Fig. 4), indicating a strong correlation. The highest $R^2$ in the middle and deep layers did not exceed 0.68 and 0.30, respectively, and the lowest was as low as 0.32 and 0.01. This shows that, as the soil layer depth increased, the correlation between air temperature and soil temperature decreased, implying that the time it took for

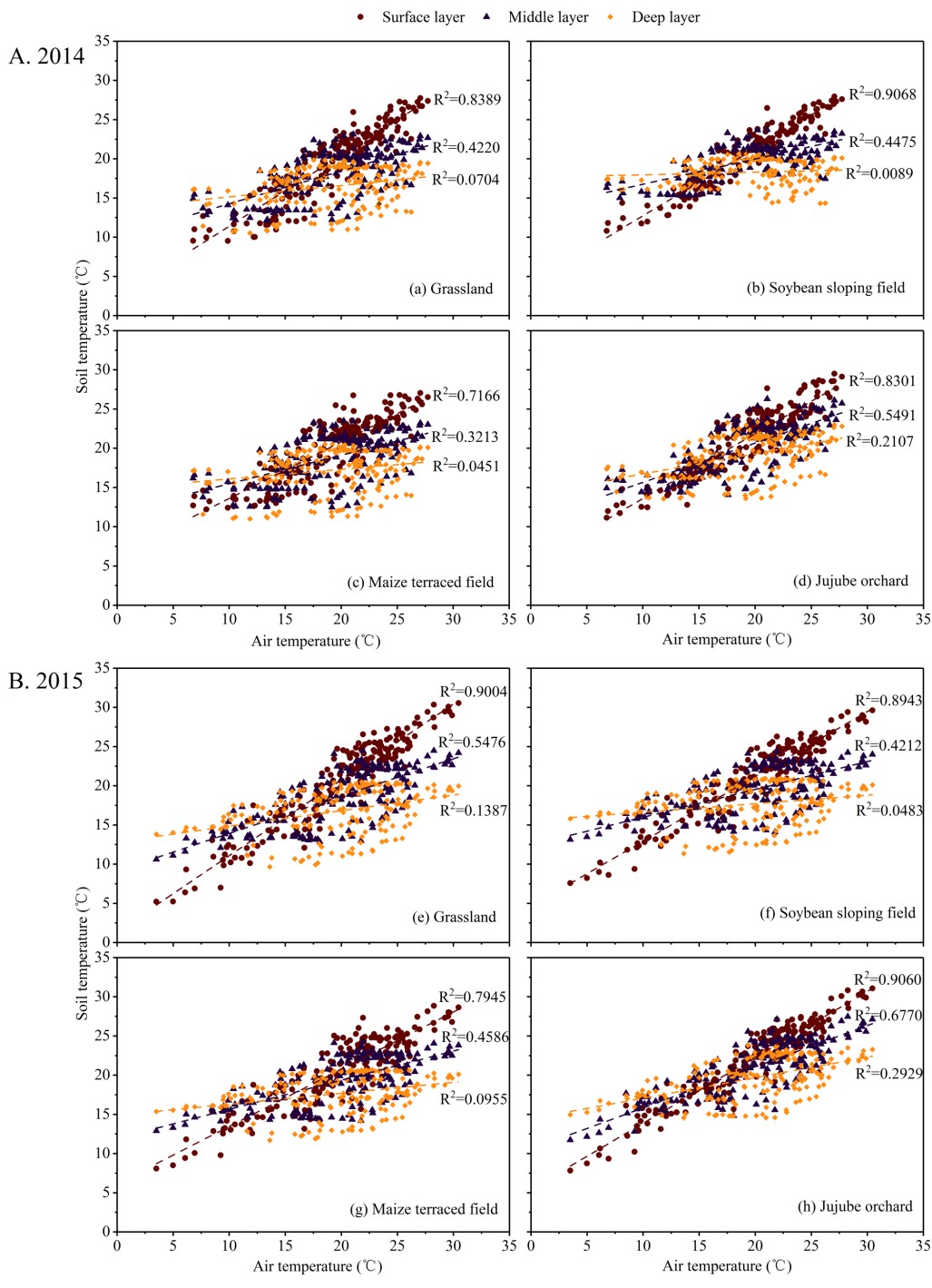

**Figure 4** **Correlations between soil temperature and air temperature at different soil depths in the experimental land-use types during the (A) 2014 and (B) 2015 growing seasons.**

the change in soil temperature to catch up to the change in air temperature increased. It is worth noting that the maximum $R^2$ between the soil and air temperature in the deep layer was less than 0.30, indicating that the deep layer's response to air temperature was rather weak. The primary cause of this phenomenon is that solar radiation heats the surface soil, which is then transferred to deep soil *via* heat conduction and convection. As the depth of the soil layer increased, the heat carried by conduction and convection decreased, causing the soil temperature to decrease. As a result, air temperature changes had a significant impact on the surface soil but not on the deep soil.

The $R^2$ between the soil temperature in the surface layer and the air temperature in the jujube orchard and grassland in the 2014 growing season was approximately 0.83, which was lower than that in the soybean sloping field, and the $R^2$ in the maize terraced field was relatively low, approximately 0.72 (Fig. 4). In the 2015 growing season, the coefficients of determination between the soil temperature in the surface layer and air temperature in the soybean sloping field, jujube orchard, and grassland were all around 0.90, which was higher than that in the maize terraced field.

## DISCUSSION

### The link between changes in soil moisture and environmental factors

The combined effects of multiple influencing factors, such as land use (vegetation, topography, *etc.*), meteorological factors (precipitation, air temperature, wind speed, *etc.*), and soil properties, result in temporal and spatial changes in soil moisture. Soil moisture retention, diffusion, and loss can all be impacted by rising air temperatures (*Qiu & Ben-Asher, 2010*; *Chen et al., 2018*). Precipitation can increase atmospheric relative humidity and soil moisture content, while solar radiation and wind speed can influence the evaporation of soil moisture. Changes in soil temperature have a significant impact on soil moisture, affecting both the maintenance and movement of soil moisture. *Cho & Choi (2014)* investigated the temporal and spatial variability of soil moisture and its relationship with meteorological factors at the regional scale of the Korean Peninsula and found that soil moisture was positively correlated with daily average precipitation but negatively correlated with air temperature. *Czarnecka & Nidzgorska-Lencewicz (2006)* found that the variability of soil moisture to a depth of 10 cm under rye and potato cultivation is mainly due to total precipitation levels. Moisture in deeper soil layers under rye depends primarily on air humidity, and under potato cultivation, on the thermal conditions of air and soil. In an oasis farmland-shelter forest, *Wang (2007)* discovered a significant negative correlation between wind speed and soil moisture, but no significant influence of solar radiation, air temperature, or atmospheric relative humidity on soil moisture content. *Han et al. (2016)* analyzed soil moisture and temperature data from four Qaidam Basin sampling sites and found that soil temperature was closely related to soil moisture, and the relationship between soil temperature and moisture at various depths at the Nomhon site was a quadratic function with a parabolic change. The relationship between soil temperature and moisture in the shallow layer was also quadratic at the Delingha and Da Qaidam sites, whereas the soil temperature and moisture in the deep

layer were positively linearly correlated. The soil moisture in the 0–60 cm soil layer under different sloping land use types responded differently to environmental factors in this study (Tables 2, 3, 4 and 5), primarily because of differences in vegetation types and coverage, as well as micro-topography, which directly affected the ground receiving precipitation and solar radiation, as well as the meteorological environment near the ground. In general, soil moisture was positively correlated with precipitation and air humidity, whereas it was negatively correlated with air temperature, wind speed, solar radiation, and soil temperature (Table 2). The results of this study differ slightly from those of previous studies, which could be due to the different regions studied (such as humid, arid, and semi-arid regions), as well as the study's scale and time (*Wang et al., 2016*; *Baldwin, Naithani & Lin, 2017*).

The direct and comprehensive effects of air humidity on soil moisture in the 0–60 cm soil layer under soybean sloping fields, maize terraced fields, and grasslands were at their peak in the 2014 growing season, according to the path analysis results (Table 3). In the jujube orchard, soil temperature in the 0–60 cm soil layer had a significant impact on soil moisture (Table 3). The direct and comprehensive effects of soil temperature on soil moisture under various sloping land use types were greatest during the 2015 growing season (Table 4). This is because increased plant transpiration and soil evaporation result from lower air humidity and higher soil temperature, resulting in a decrease in the soil moisture content (*Li et al., 2002*; *Kidron & Kronenfeld, 2016*). The environmental variables entered for the four sloping land use types in the 2014 and 2015 growing seasons were air temperature, air humidity, solar radiation, wind speed, and soil temperature in the 0–60 cm soil layer, according to the stepwise regression analysis in this study (Table 5). The above results differ from the findings of *Zhang et al. (2013)*; that is, the daily mean air temperature was included in the input variable in this study, while it was excluded from Zhang et al.'s research results and did not enter the input variable. The difference in input variables is due to the difference in study area and time, and the direct effect of air temperature on soil moisture in Zhang et al.'s study was small, with a direct path coefficient of only −0.0364, and the comprehensive determination ability of air temperature on soil moisture was small. The air temperature was entered into the regression model of the soil moisture under the maize terraced field and jujube orchard in this study during the 2014 and 2015 growing seasons (Table 5), especially in the dry year, where the direct path coefficient between the air temperature and the soil moisture under the maize terraced field and jujube orchard was both greater than 0.54 (Table 4). Although the indirect effect of air temperature on soil moisture through other environmental factors (air humidity, wind speed, and soil temperature) was the opposite of the direct effect, the combined effect of air temperature on soil moisture under these two sloping land use types was still relatively large (Table 4). This may be because the soil in the jujube orchard is exposed over a large area, and the change in soil moisture is mainly affected by evapotranspiration, while the change in air temperature changes the atmospheric evaporative force, resulting in changes in soil evaporation and jujube transpiration (*Li et al., 2018*). Compared with the other three sloping land use types, terraced fields reduce the ground slope, change the original topography, flatten the field surface, avoid the generation of runoff, and store rainwater, increase infiltration, and improve soil moisture through interception, which

has been confirmed in previous studies by the authors and others (*Tang et al., 2019a*; *Tang et al., 2019b*; *Xu et al., 2021*). High soil moisture content implies that there is sufficient water in the soil for evaporation, and evaporation changes are mainly affected by air temperature changes (*Kohfahl, Saaltink & Bermudo, 2021*). Therefore, soil moisture in the maize terraced field was closely related to air temperature. Air temperature did not enter the soil moisture regression model of soybean sloping fields and grasslands in the 2014 and 2015 growing seasons (Table 5), indicating that the effect of air temperature on soil moisture was weak and almost negligible. This may be because the leaf area index and coverage of soybean and natural grass are higher than those of jujube orchards with large areas of bare soil, and the vegetation covers the soil surface intensively, which hinders the exchange of water vapor and heat in the soil with the outside air and weakens the response of soil moisture change to air temperature. In addition, the sum of the indirect path coefficients of solar radiation under the soybean sloping field, maize terraced field, and grassland in the 2014 growing season was high, with values of $-0.442$, $-0.467$, and $-0.581$, respectively (Table 3). The absolute value of the indirect path coefficient of solar radiation affecting soil moisture through air temperature, air humidity, and soil temperature was above 0.384, and the maximum value was as high as 0.603 (Table 3), indicating that solar radiation had a substantial impact on soil moisture through air temperature, air humidity, and soil temperature, according to the composition of the indirect path coefficient of solar radiation.

## Soil temperature changes in response to environmental factors

Numerous studies have shown that air temperature is the most important meteorological factor that affects soil temperature (*Paul et al., 2004*; *Wan et al., 2007*). The grey relational grade between air temperature and soil temperature in the 0–60 cm soil layer under the four sloping land use types was found to be the highest in this study, ranging from 0.8275 to 0.8446 (Table 6), indicating that air temperature was the primary factor affecting the soil temperature of the 0–60 cm soil layer under different sloping land use types. The grey relational grade between soil and air temperature decreased as the soil depth increased (Table 6), indicating that the influence of soil temperature on air temperature weakened as the soil depth increased. As a result, during the 2014 and 2015 growing seasons, air temperature was not the primary driver of soil temperature change in the 60–100 cm soil layer under soybean sloping fields, maize terraced fields, and grasslands. However, in the 60–100 cm soil layer of the jujube orchard, air temperature was still the most important factor influencing soil temperature, and it could be because the jujube orchard is exposed and the vegetation coverage is low, resulting in poor ground shading. However, the soil moisture content in the 60–100 cm soil layer of the jujube orchard was low, and many pores in the soil were filled with air. Water has a specific heat capacity of approximately three times that of air, which means that the temperature of water rising or falling is approximately one-third that of air when absorbing or releasing the same heat. As a result, the lower the soil moisture content, the weaker the regulating effect on soil temperature and the easier it is for external meteorological conditions to affect soil temperature. Owing

to the aforementioned two factors, air temperature has a significant impact on the deep soil of the jujube orchard.

The coefficient of determination between soil temperature and air temperature in different soil layers of 0–100 cm under the four sloping land use types was different in the study of the response of soil temperature to air temperature (Fig. 4), and the coefficient of determination in the surface layer under the maize terraced field was small (Figs. 4C–4G), indicating that the response was weak. This could be because the plants in soybean sloping fields and grasslands are short, whereas maize plants in terraced fields are tall, and the densely covered maize canopy can intercept a large amount of solar radiation. At the same time, owing to low chlorophyll concentration and a waxy layer on the surface of the leaves, maize leaves have a higher reflectivity than soybean leaves, according to *Liu et al. (2012b)*. In addition, the author's previous research found that the water storage and moisture retention effects of terraced fields were quite significant compared with sloping fields; therefore, the soil moisture content in the 0–20 cm soil layer of the maize terraced field was higher than that of the other three sloping land use types in the 2014 and 2015 growing seasons (*Tang et al., 2019a*). Water has a much higher specific heat capacity ($4.2 \times 10^3$ J/(kg °C)) than soil ($1 \times 10^3$–$2.5 \times 10^3$ J/(kg °C)), which means that the temperature of water rises less than that of soil when they absorb the same amount of heat, whereas the ability of water to lose heat during the cooling process is lower than that of soil. Therefore, the soil specific heat capacity increases as soil moisture content increases, and the higher the soil moisture content, the slower the soil temperature rises and falls, and the less sensitive it is to air temperature. Based on the above results and considering the fact that the loess hilly region has many sloping fields, a wide area of dry land, and low precipitation utilization, it is encouraged to transform sloping fields into terraced fields, strengthen the construction of terraced fields, promote water storage and moisture conservation, enhance soil and water conservation capacity, and alleviate drought and soil erosion. Therefore, it is of great significance to promote the sustainable development of agricultural production in arid areas.

## CONCLUSIONS

(1) Air temperature and humidity were the main factors affecting soil moisture changes in soybean sloping fields and grasslands during the 2014 and 2015 growing seasons. The most important meteorological factors affecting changes in soil moisture in the maize terraced field were air humidity and wind speed. In the jujube orchard, the effect of air temperature on soil moisture was significant ($p < 0.01$) and negative. For all sloping land use types, there was a negative correlation between soil moisture and soil temperature in the 0–60 cm soil layer ($p < 0.01$). Air humidity had the greatest direct and comprehensive impact on soil moisture in soybean sloping fields, maize terraced fields, and grasslands during the growing season of the normal precipitation year (2014). Soil temperature had a relatively large effect on soil moisture in jujube orchards, and the impact on soil moisture in soybean sloping fields and grasslands was mostly reflected in the direct effect. The direct effects of air temperature, air humidity, and wind speed on soil moisture in the maize terraced field were

predominant. The direct and comprehensive effects of soil temperature on soil moisture of various sloping land use types were the largest and showed a negative effect during the growing season of the dry year. (2) In the four sloping land use types, air temperature had a strong correlation with the soil temperature of the 0–60 cm soil layer during the 2014 and 2015 growing seasons, especially in the 0–20 cm soil layer. The correlation was the highest, and the grey relational grade and correlation both decreased as soil depth increased. The soil temperature in the 0–20 cm soil layer under the maize terraced field had the weakest response to air temperature. Air temperature was the most important factor affecting soil temperature in the 60–100 cm soil layer of the jujube orchard. (3) The maize terraced field had favorable effect on soil moisture conservation and soil temperature regulation; therefore, it is encouraged to transform sloping fields into terraced fields in the loess hilly region to improve the efficient use of natural precipitation and promote the sustainable use and improvement of sloping fields.

## ACKNOWLEDGEMENTS

The authors are grateful to Drs. Lusheng Li and Wenhao Sun for their kind help in the installation of experimental equipment and data acquisition.

### Funding

This work was supported by the Natural Science Research General Program of Jiangsu Higher Education Institutions (No. 21KJB210022), the ''Lv Yang Jin Feng'' Program of Yangzhou (No. YZLYJF2020PHD088), the National Natural Science Foundation of China (No. 51909228), the China Postdoctoral Science Foundation (No. 2020M671623), and the Natural Science Foundation of Jiangsu Province (No. BK20210824). The funders had no role in study design, data collection and analysis, decision to publish, or preparation of the manuscript.

### Grant Disclosures

The following grant information was disclosed by the authors:
Natural Science Research General Program of Jiangsu Higher Education Institutions: 21KJB210022.
''Lv Yang Jin Feng'' Program of Yangzhou: YZLYJF2020PHD088.
National Natural Science Foundation of China: 51909228.
China Postdoctoral Science Foundation: 2020M671623.
Natural Science Foundation of Jiangsu Province: BK20210824.

### Competing Interests

The authors declare there are no competing interests.

## Author Contributions

- Min Tang conceived and designed the experiments, performed the experiments, analyzed the data, prepared figures and/or tables, authored or reviewed drafts of the article, and approved the final draft.
- Wanning Li analyzed the data, prepared figures and/or tables, and approved the final draft.
- Xiaodong Gao conceived and designed the experiments, authored or reviewed drafts of the article, and approved the final draft.
- Pute Wu conceived and designed the experiments, authored or reviewed drafts of the article, and approved the final draft.
- Hongchen Li performed the experiments, prepared figures and/or tables, and approved the final draft.
- Qiang Ling performed the experiments, authored or reviewed drafts of the article, and approved the final draft.
- Chao Zhang analyzed the data, prepared figures and/or tables, authored or reviewed drafts of the article, and approved the final draft.

## Field Study Permissions

The following information was supplied relating to field study approvals (*i.e.*, approving body and any reference numbers):

Heping Liu rented the field where the experiments were conducted.

## Data Availability

The raw data are available in the Supplemental Files.

## Supplemental Information

Supplemental information for this article can be found online at http://dx.doi.org/10.7717/peerj.13736#supplemental-information.

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
