# Peer review of "Land use affects the response of soil moisture and soil temperature to environmental factors in the loess hilly region of China"

_PeerJ, doi:10.7717/peerj.13736_

## Round 0.1 · original submission · Major Revisions

Dear authors,

As you shall see, out of the three reviewers, reviewer 1 has raised serious issues regarding your manuscript. Whereas reviewer 3 has also pointed out various serious issues. However, he has also provided some guidance. I have decided to go for Major Review, considering the fact, you shall be able to defend your work and incorporate the suggestions provided by all three reviewers. In this regard, while you upload the revised manuscript please make sure to upload point-by-point responses to all the reviewers' comments. Further, the revised manuscript must highlight the segments/ sentences/ paragraphs/data/analysis that has been revised.

Reviewer 1 ·

Basic reporting

The language can be improved, but some words used need to be definitely replaced. Literature references are not sufficient, excluding the introduction section. There is no background provided in the introduction section on the four crop types and similar studies for these. The article structure can be improved. More figures are required to graphically show the patterns of the variation of the soil moisture, maps for regional topography, geography, and geology.

Experimental design

The manuscript matches the aim and scope of the journal but lacks in providing significant information. The research gaps are ill-explained and the utilization of the solutions is not discussed. The investigation falls short due to several misses in parameter selection. Methods are rudimentary and discussed in sufficient detail.

Validity of the findings

This work lacks significant novelty, and the impact is uncertain, as there are issues with the design of the study and in the discussion of the sources of error and uncertainty.

Additional comments

The authors investigated the inter-relationship between different meteorological parameters and soil moisture for four different types of crops in a hilly region. The objectives of the manuscript are weak, there are some missing ends not considered in the analysis, the study itself is fairly rudimentary and the write-up is poor. I reject the publication of this manuscript on these grounds. Detailed comments are as follows.
• The introduction section doesn’t discuss the four different types of crops for which the investigation is conducted.
• Line 121: “Situation”?
• There is no map of the study area or Figure illustrating the geographical location, the topography, and the geology of the area.
• The topography is not discussed at all.
• Please discuss the specifications of the two sensors used, particularly their precision. References should be cited which indicate the application of these sensors in previous studies. There is no discussion on why these were used and what advantages these have against conventional sensors such as the hydra/theta probes.
• A map of the area with the spatial distribution of the different crops would be appreciated by the readers in this manuscript.
• The referencing is insufficient except in the introduction section.
• The impact of slope-aspect is missing.
• The impact of soil properties is missing.
• The impact of surface and soil temperature is missing.

Reviewer 2 ·

Basic reporting

the basic reprting is clear and easy to read. but the authors did not provide some erferences in old days. It is very importent for use to use the references which publishied at the begining of this research field.

language is not concise and clear in some paragraphs.

Experimental design

if possible, please add one or two photographes of experiment landuses .

Validity of the findings

no comment

Additional comments

soil moisture could be deeply affected by envrinment factos in semi- dry land. it is well studied since last century. based on the new techniques in soil moisture and air teperature survy, the landuses affects on the response of soil moisture and soil teperature to environment factors can be precisly studied. this paper has done the field experiment and comprehensive analysis to reveal the linkage of land use and soil moisture and surronding environment. the paper did not provide enough new foundinds in the mechanism, but provise very sound presentation in the reseach area.
Your introduction needs more detail, such as the basic foundings in last century. I suggest that you improve the description at lines 44- 69 to provide more justification for your study.
for table 1, you may need to add the real meaning of X1--X6, give a note below to interpret the letters stead for.
L136 to L140 is not clear in meaning. L150-151 IS NOT CLEAR that the "10 -meter " here.
you need to give the reason why you choose the 4 types of land uses. what is the difference among landuses. such as the crop height and leaf area index. eigher in the experiment design or in the discussion section.
L510 -515, I am not understand your meaning. please make the sentence much clear. do you mean that Zhang' research did not input the aie teperature in Zhang's study.

Annotated reviews are not available for download in order to protect the identity of reviewers who chose to remain anonymous.

·

Basic reporting

Thank you for submitting your manuscript on Land use affects the response of soil moisture and soil
temperature to environmental factors in the loess hilly region of China. With the changing climactic conditions globally soil moisture is a very important parameter in determining the productivity of land for sustenance of food security. The authors have touched a very important aspect in this regard. The manuscript needs a lot of improvement in text such as introduction and conclusion. Literature review is sufficient however too old references may be avoided. The background and the context is relevant though however too lengthy. The structure and flow of the article is good and the figures/tables are self explanatory. The authors set the goal of this research is to uncover the mechanisms by which various environmental factors influence the soil moisture.

Therefore, I giving some suggestions which I hope useful to the author.

Experimental design

The experiment design is good however lacks novelty, the authors may explain how their methods are different from other researchers. Although, lot of effort has gone into the research but the research questions are not well defined. The research fills the knowledge gap but authors have not clearly expressed the same.

Validity of the findings

All the data has been provided and is supported by robust statistical analysis and tests, however validation of results has not been performed.
Although the authors set the goal of this research is to uncover the mechanisms by which various environmental factors influence the soil moisture, a section has been dedicated for the same however it discussed the relations only I could not find any explanation for "mechanism".

The conclusion again repeats the detailed results. The conclusion must be crisp and should summarize the results with applicability of results and methods globally, it should highlight limitations and recommendations inferred from the results.

Additional comments

Although the research is well thought and has applicability in current food security scenarios, the paper needs major improvements in defining the goals and achieving them and then explaining them. The authors may explain the relevance of their research in terms of its impact on global food security and sustainable development goals. Discussion and Conclusion needs improvement. English language needs to be lucid and understandable, although it is but at some places its giving misleading meanings. I have highlighted few of them. The authors may check entire manuscript for such rephrasing.

---

## Round 0.2 · accepted · Accept

Thank you very much for responding to each comment by all the reviewers. The manuscript is well structured now.